# Detecting joint attention events in mother-infant dyads: Sharing looks cannot be reliably identified by naïve third-party observers

**Kirsty E. Graham**[1,2☯]*, **Joanna C. Buryn-Weitzel**[1‡], **Nicole J. Lahiff**[1‡], **Claudia Wilke**[1‡], **Katie E. Slocombe**[1☯]

**1** Department of Psychology, University of York, York, United Kingdom, **2** School of Psychology & Neuroscience, University of St Andrews, St Andrews, United Kingdom

☯ These authors contributed equally to this work.
‡ JCB-W, NJL and CW also contributed equally to this work.
* keg4@st-andrews.ac.uk

**Data Availability Statement:** All relevant data are within the manuscript and its Supporting Information files.

**Funding:** This study was funded by an European Research Council Consolidator grant (ERC_CoG

## Abstract

Joint attention, or sharing attention with another individual about an object or event, is a critical behaviour that emerges in pre-linguistic infants and predicts later language abilities. Given its importance, it is perhaps surprising that there is no consensus on how to measure joint attention in prelinguistic infants. A rigorous definition proposed by Siposova & Carpenter (2019) requires the infant and partner to gaze alternate between an object and each other (coordination of attention) and exchange communicative signals (explicit acknowledgement of jointly sharing attention). However, Hobson and Hobson (2007) proposed that the quality of gaze between individuals is, in itself, a sufficient communicative signal that demonstrates sharing of attention. They proposed that observers can reliably distinguish "sharing", "checking", and "orienting" looks, but the empirical basis for this claim is limited as their study focussed on two raters examining looks from 11-year-old children. Here, we analysed categorisations made by 32 naïve raters of 60 infant looks to their mothers, to examine whether they could be reliably distinguished according to Hobson and Hobson's definitions. Raters had overall low agreement and only in 3 out of 26 cases did a significant majority of the raters agree with the judgement of the mother who had received the look. For the looks that raters did agree on at above chance levels, look duration and the overall communication rate of the mother were identified as cues that raters may have relied upon. In our experiment, naïve third party observers could not reliably determine the type of look infants gave to their mothers, which indicates that subjective judgements of types of look should not be used to identify mutual awareness of sharing attention in infants. Instead, we advocate the use of objective behaviour measurement to infer that interactants know they are 'jointly' attending to an object or event, and believe this will be a crucial step in understanding the ontogenetic and evolutionary origins of joint attention.

2016_724608) awarded to KS (https://erc.europa.
eu/funding/consolidator-grants). The funders had
no role in study design, data collection and
analysis, decision to publish, or preparation of the
manuscript.

**Competing interests:** The authors have declared
that no competing interests exist.

## Introduction

The emergence of joint attention, the ability of two individuals to share attention about an external object or event, is considered a crucial step in child development [1]. Joint attention is thought to emerge in infancy at around 9–12 months [1–3], and may be important in language acquisition, with children connecting words with objects to which they and another individual are jointly attending [1, 4, 5]. Some have also argued that engaging in joint attention lays the foundation for shared intentionality, which in turn underpins human cooperation [6]. Given the links between engaging in joint attention and both language and cooperation, it has been suggested that joint attention might represent a key species-difference between humans and other great apes [7]. However, the extent to which non-humans engage in joint attention is fiercely debated, with some studies suggesting non-human great apes have joint attention skills [8–10] and some studies finding no evidence for engagement in joint attention events [3]. This debate is currently hampered by the lack of a consistent and clear definition of joint attention events that can be applied comparatively across species.

Given the developmental and evolutionary importance of joint attention, it is perhaps surprising that there is a lack of consensus in the literature as to both what constitutes joint attention and how to operationalize definitions. Most studies focus on joint attention skills, and examine an individual's ability to initiate joint attention with a partner or respond to joint attention cues, such as the gaze or pointing gestures of a partner. Joint attention skills are necessary but not sufficient for engaging in joint attention events, which also require an individual to have the motivation and opportunity to use those skills with willing partners. What conceptually constitutes a full joint attention event, where at least two people actually engage in joint attention, remains diverse. In the literature, a 'Joint attention event' can refer to behaviours as varied as spectators in a stadium all watching the same football match, and a child following the gaze of their mother to jointly attend to a butterfly and exchanging emotional reactions to the butterfly, making the jointness of the interaction manifest [11]. Similarly, when it comes to operationalizing joint attention events, a diverse array of approaches also exists (Table 1). In order to better understand this variation, we conducted a systematic search for operational definitions of joint attention events published in English in journal articles from 2000–2018 by inserting 'Joint Attention' into Science Direct, and then manually checking to remove papers that do not define and measure joint attention events, or that only define initiating joint attention skills (IJA) or responding to joint attention skills (RJA). We also excluded studies that did not examine human-human interactions, e.g. human-video, human-robot, nonhuman-human etc. (Table 1).

These definitions of joint attention represent a wide range of behaviours and interactions with varying degrees of 'jointness'. At the simplest end, it is sufficient for two individuals to both look at the same object (also called 'parallel attention' [39]), or for two individuals to look at one another; and at the more complex end, individuals must coordinate their attention between an object or event and one another and also communicate with one another about it (Table 1). The problem inherent in the simpler operational definitions of joint attention events is that such behavioural patterns could be explained by (a) properties of the event itself (a salient event, such as a loud noise, may draw the attention of multiple individuals, but they might not be aware of others also orienting to the same event) or (b) by the desire to check or monitor the behaviour of another (an event may be of interest but so is monitoring the behaviour of another individual, resulting in gaze alternation between these competing loci of attention). Whilst monitoring or checking the behaviour of another individual might ultimately have sharing motives, one might also engage in this behaviour for more individualistic or competitive reasons (e.g. an infant may want to check the mother is remaining close-by or an

**Table 1. A list of operationalized definitions of joint attention events used in papers between 2000 and 2018 that met criteria for our literature review; references [12–104].**

| Operationalized Definition for Joint Attention events | Studies that use this definition |
|---|---|
| Attend to partner's actions | Gonsiorowski et al. 2016 |
| Both attend to an object/event at the same time (*no mutual gaze required*) | Arens, Cress & Marvin 2005*; Cress, Arens, & Zajicek 2007*; Craig et al 2004; Hughes & Allen 2013; Kompatsiari et al 2018; Macdonald & Tatler 2018; Najnin & Banerjee 2018; Povis & Crowley 2015; Richardson et al 2007; Sung & Hsu 2009; Walberg & Craig-Unkefer 2010; Yont, Snow & Vernon-Feagans 2003; Yu & Smith 2013; Yu & Smith 2016; Yu & Smith 2017 |
| Mutual Gaze (*no attention to object required*) | Charman et al. 2000*; Hobson & Hobson 2007**; Morgan, Maybery & Durkin 2003; Sanefuji et al. 2009 |
| Gaze switch: object-partner **OR** partner-object | Aarne & Tallberg 2010; Arens, Cress & Marvin 2005*; Charman et al. 2000*; Charman et al. 2003; Cress, Arens, & Zajicek 2007*; Clifford & Dissanayake 2009; Vismara et al. 2018; Yu & Ballard 2007 |
| Gaze switch + partner communication (*i.e. individual switches gaze to an object while partner is communicating about that object*) | Matatyaho & Gogate 2008 |
| Gaze alternation: object-partner-object **OR** partner-object-partner | Arens, Cress & Marvin 2005*; Clifford & Dissanayake 2008; Cress, Arens, & Zajicek 2007*; Spector & Charlop 2018; Striano & Bertin 2005 |
| Coordinates attention between partner and object/event (*this involves looking at both the partner and the object/event but does not specify order, frequency, or duration*) | Aldrich et al 2015; Arnold et al. 2000; Brune & Woodward 2007; Cleveland & Striano 2007; Conboy et al. 2015; Gaffan et al. 2010; Gulsrud et al. 2007; Grossmann & Johnson 2010; Johnson et al. 2008; Legerstee, Markova & Fisher 2007; Neerinckx et al 2014; Nelson, Adamson, & Bakeman 2008; Nordahl-Hansen et al. 2016; Saxon et al. 2000; Striano & Bertin 2005; Striano et al. 2009; Suma et al. 2016; Trautman & Rosenthal 2006; Yazbek & D'Entremont 2006 |
| Coordinates attention "for the purpose of sharing" | Ingersoll & Schreibman 2006 |
| Coordinates attention between partner and object/event + acknowledges partner's participation (quality of looks may be used to judge acknowledgement of partner's participation) | Adamson et al. 2009; Adamson et al. 2010; Bigelow 2003; Childers, Vaughan, & Burquest 2007; Gauthier et al. 2011; Gulsrud et al. 2016; Hahn et al. 2016; Kasari et al. 2010; Larkin et al. 2015; Markus et al. 2000; Mastin & Vogt 2016; Rice et al. 2016; Schechter et al. 2010; Schertz & Odom 2004; Schertz et al. 2018; Smith et al. 2009 |
| *Early Sociocognitive Battery* (ESB): alternating gaze between partner and object | Roy & Chiat 2014 |
| Attends to an object/event + [gazes at partner **AND/OR** communicates with partner] | Adamson, Deckner, & Bakeman 2010; Allely et al. 2013; Benigno & Farrar 2012; Benigno et al. 2007; Bigelow, Maclean & Proctor 2004; Bigelow et al. 2010; Christidou 2018; Depowski et al 2015; de la Ossa & Gauvain 2001; Henderson & Jennings 2003; Ine, Heleen & Bea 2011; Morales et al. 2005; Nowakowski, Tasker & Schmidt 2012; Nowakowski et al. 2009; Nowakowski et al. 2011; Pierce et al. 2015; Shire et al. 2016; Slaughter et al. 2008; Skarabela 2007; Skarabela & Allen 2010; Tasker & Schmidt 2008; Tasker, Nowakowski, & Schmidt 2010; Yoon et al. 2014; Zampini, Salvi & D'Odorico 2015 |
| Coordinates attention between partner and object/event + acknowledges partner's participation + communication with partner | Chiang et al. 2016; Ketelaar et al. 2012; Landry et al. 2008 |

*Papers that use more than one definition within the paper

**Papers that specify types of look

individual may want to check that another individual is not taking a valuable food item). It is also important to note that simpler forms of 'joint attention' such as the simultaneous monitoring of an object or event by multiple observers emerge earlier in development (common in 6-month olds (39)) than more complex forms of joint attention, supporting the idea that different cognitive processes may underlie some behaviours currently labelled as 'joint attention' in the literature.

To try and capture the 'jointness' of a joint attention event, more recent operational definitions seek to identify behavioural markers of mutual awareness that the interaction partners are attending to the event together and are thus sharing attention. One such behavioural marker is communication, which is a necessary element of the 'more complex' operational definitions in Table 1. Whilst facial expressions, such as smiles or looks of concern, and vocalisations are forms of communication that can be objectively and reliably coded from interaction partners, it has also been suggested that the quality of looks exchanged during mutual gaze can be sufficient to communicate awareness of the shared nature of the engagement (47). This suggestion is implicit in the definitions that include "acknowledges partner's participation" (Table 1; e.g. (13), p. 19), which is often assessed by mutual gaze although there is often little description given as to what it is about the gaze that acknowledges the partner's participation. Supporting this argument, and providing further detail on how to assess gaze, Hobson and Hobson assert that observers can reliably identify 'sharing looks' in children and distinguish them from checking or orienting looks [47]. In this study, two raters classified the looks of 11-year-old children with and without autism towards an experimenter into three categories according to the definitions detailed in Table 2, and significant agreement between the two

**Table 2. Definitions of types of look from Hobson & Hobson, 2007 and the summaries of these definitions provided to participants in the current study.**

| Type of Look | Definitions from Hobson & Hobson (2007) | Definitions adapted for participant instructions in the current study |
|---|---|---|
| Sharing | "Sharing looks" were defined as those looks directed to the tester that could be seen to express a participant sharing experience through interpersonal contact with the tester. They involved a deep gaze that conveyed personal involvement . . . in contrast to checking looks that involved glances "at" the tester. | **Sharing look**: The child looked into the mother's eyes and the look conveys personal involvement and emotional contact. The child's look acknowledges the mother's gaze (reciprocal) |
| | • *Reciprocal*: The participant appears to register that the tester is also looking to the participant. | |
| | • *Deep*: The look is *into* the eyes of the tester. | |
| | • *Personal*: The look manifests affective contact with the tester. | |
| Checking | "Checking looks" were defined as those looks toward the tester that were used to assess or check out either the situation or the tester's response. | **Checking look**: The child looked at the mother's face and eyes, likely to monitor or check on the mother's presence and behaviour, maybe to help the child assess the situation. The child's look does not acknowledge the mother's gaze (not reciprocal). |
| | • *Nonreciprocal*: The participant appears to look to the tester without registering the tester's look to the participant. | |
| | • *Superficial*: The look is at the eyes of the tester. | |
| | • *Impersonal*: The look appears to have the goal of monitoring the tester's actions. | |
| Orienting | "Orientating looks" were those that appeared to occur in direct response to an action, sound, or movement on the part of the tester. | **Orienting look**: The child looked at the mother as a direct response to an action, sound (including language) or movement the mother made (the mother may or may not have been intending to gain their attention) |

raters was found. It is unclear, however, whether the looks of preverbal infants can be reliably classified in the same way and it is also important to test this with a greater number of raters.

Here, we aim to robustly test whether raters can accurately judge the type of look given by an infant to their mother as sharing, checking or orienting looks. This will enable us to evaluate whether the type of look should be considered a valid form of communication upon which judgements can be made about joint attention events occurring between preverbal infants and their mothers. We asked a group of 32 naïve raters to code infant looks to their mothers during free play according to the Hobson & Hobson definitions (Table 2). Raters judged the type of look in 63 short video clips cut 3 seconds before and after the infant looks at the mother, and also gave a rating of confidence in their judgement. For 33 videos, we had the mother's judgement as to the type of look her infant gave to her. We expected that if the types of looks could be reliably identified by naïve observers then we would obtain high levels of agreement between (i) our 32 raters, and (ii) our raters and the mothers. We aimed to explore whether self-rated confidence was associated with high levels of agreement with other raters. For looks where there was strong agreement between raters, we then performed analysis of the video to try and ascertain which behavioural cues the raters may have been basing their judgements on.

## Methods

### Ethics statement

The University of York, Department of Psychology, Departmental Ethics Committee approved the study (ID number 605). Written consent was obtained from all participants.

### Participants

We recruited 32 women in York, UK, including 14 mothers and 18 non-mothers. This allowed us to test whether experience raising infants affected agreement on the categorisation of infant looks. The average age for the mother group was 45.21 years (SD±8.32) and for the non-mother group was 32 years (SD±8.86). An independent-samples t-test revealed that the mother group were significantly older than the non-mother group (t(29) = 4.2, p<0.001).

### Stimuli and experimental design

Stimuli consisted of videos that were cut into short clips (mean = 8.11s, SD = 1.70s) that started 3s before the child looked at their mother and ended 3s after the child stopped looking at their mother, unless there was another look within that 3s in which case it was cut immediately before or after the second look. This time window was chosen to give raters an opportunity to observe part of the interaction in which the look occurred, but without providing broader context and behaviours, as we wanted raters to judge the quality of the look, not the general intent of the infant or mother. We presented participants with two sets of videos: "Set 1" contained 33 videos filmed in 2017, and "Set 2" contained 30 videos filmed in 2009–2010. For Set 1, the experimenters had filmed 7 mothers and infants living in or near York, UK, playing (age: mean = 15.75 months, min = 8 months, max = 26 months). We had interviewed the mothers immediately afterwards, showing them the videos and asking them to state what type of look they thought the infant had given them, using the same criteria as participants received (Table 2), with the exception that we also allowed mothers to also say if they didn't know ("Unknown"). This meant that the mothers were relatively confident in the sharing, checking and orienting look categorisations they provided. We selected groups of three videos for each mother-infant dyad, containing one of each type of look where possible (as judged by the mother). Only four looks in Set 1 were classified as "Unknown" by the mother. Three mother-

infant dyads provided one group of three looks and four dyads provided two groups of three looks. The Set 2 videos were taken from 8 mothers and infants living in or near York, UK (age: mean = 11.125 months, min = 11 months, max = 12 months) and we had no judgements from these mothers on the type of look the infant produced. We selected groups of three high quality videos for each infant. Six mother-infant dyads provided one group of three looks and two dyads provided two groups of three looks.

The experiment was presented to participants using an online platform (Gorilla.sc), which was set to "private" so that only the experimenters had access. Video dimensions for Set 1 were 16:9 with size of 21.3x12cm; Set 2 were 4:3 with 16x12cm. Within both sets of videos, the videos were shown in blocks of three that contained videos from the same mother and infant dyad. Videos were randomised within the blocks, and the order of blocks was randomised across the experiment. We also counterbalanced the order of the response buttons (three types of look) across participants. There were six possible orders for the three response buttons (sharing, checking, orienting) and participants were randomly assigned to one of these six orders. Participants were given a printed sheet of the definitions of the three types of looks (Table 2) to refer to throughout the experiment and we ensured that the order of definitions on the sheet matched the order of response buttons in the experiment.

## Procedure

The experimenter started the experiment for the participant, who was seated in a quiet room in front of a laptop computer (HP Elitebook, 14" display, 64-bit operating system). The participant was first provided with instructions on the screen, which informed them that the videos would play only once and if they saw the look they should choose one of the three types of looks (sharing, checking, orienting) even if they were uncertain, but if they did not see the look from the infant they should select the 'did not see look' button. We wanted to both standardise the number of times participants could view a video and to reduce the effects of fatigue on performance. Therefore, the video played only once in each trial. When the participant had selected the type of look, on the next screen they were asked to rate how confident they were in their judgement on a scale of 0–10 (0 = "Not at all confident"; 10 = "Extremely confident") using a continuous slider. The confidence marker's starting position on all trials was 1. There was no time limit on the duration participants spent making their judgement. Once participants had completed the confidence rating, they pressed a button that read "Next" to start the next trial, so the experiment progressed at a pace chosen by the participant.

## Video coding

To assess the possible cues that raters may have been relying on to make their judgements, we measured the duration of infant looking to mother, presence and duration of mutual gaze between infant and mother, and infant or mother communication (frequency of vocalisations, gestures or salient facial expressions during (i) the infant's look to the mother and (ii) the whole video clip). Videos were coded using The Observer XT 14 software, and the following measures were extracted: infant looking direction (mother's face, elsewhere), mother looking direction (infant's face, elsewhere), and infant and mother facial expressions, gestures, and vocal communication (including vocalisations and language). All changes in the looking direction category, no matter how short were coded. The three types of communication were coded as a new instance when they changed to a new signal within the same communication type (e.g. from a smile to a frown) or when there was a gap of >1s between signals (e.g. talking, 1.2s pause, talking). Instances of each communication type were considered separately, so that a simultaneous smile and talking represent one facial expression and one vocal communication.

To assess the reliability of the video coding, KG coded all videos and CW coded 6 videos from Set 1 and 6 videos from Set 2 (20% of all videos). Using the reliability function in Observer software, we ascertained the two video coders had high levels of agreement: duration of (i) looks from the infant to their mother's face (kappa = 0.88) and (ii) mutual gaze between infant and mother (occurred in 4/12 videos; kappa = 0.98); Number of infant facial expressions, gestures, and vocalizations (kappa = 0.78); and number of mother facial expressions, gestures, and vocalizations (kappa = 0.76).

### Data analysis

It was necessary to make some exclusions from the data set prior to analysis: There were two videos where >3 participants did not see the look (n = 16, 20) and we excluded these videos entirely from analyses. There was also one video that was accidentally duplicated, and we removed the second appearance of this video from analysis. These exclusions left 30 videos for analysis in Set 1 and 30 videos for analysis in Set 2.

As the number of instances of communication produced by infants and mothers in these short video clips were low, we extracted the following measures from the coded videos: (i) during the infant's look to the mother we considered the presence/absence of each type of communication (facial, gestural, vocal) for both the infant and mother and (ii) during the whole video clip we considered the total number of communication events (sum of all vocal, gestural and facial communication events coded) produced by (a) the mother and (b) the infant.

All analyses were conducted in R 3.5.3 [105], and we used the packages "irr" for Fleiss' kappa [106], "lme4" with "lmerTest" for GLMMs [107, 108], and "ggplot2" with "plyr" for plotting [109, 110]. Descriptions of each model are included in the relevant results sections.

## Results

### Overall agreement among naïve raters

First, we calculated Fleiss' kappa to check overall inter-rater reliability for both sets of videos [111]. For Set 1 with 30 videos and 32 raters, Fleiss' kappa was 0.157 (z = 28.4). For Set 2 with 30 videos and 32 raters, Fleiss' kappa was 0.228 (z = 40.1). These are both low rates of overall agreement [112], suggesting that in general, participants did not agree on types of look.

We asked our 32 raters whether they had children (n = 14) or did not have children (n = 18), as we thought that experience with young children might affect their judgements. For Set 1, raters who were mothers had a kappa of 0.184 (z = 14.1), and raters who were not mothers had a kappa of 0.155 (z = 15.6). For Set 2, raters who were mothers had a kappa of 0.234 (z = 17.7), and raters who were not mothers had a kappa of 0.226 (z = 22). Descriptively, mothers and non-mothers seem to agree on the type of looks at comparable rates, so for all further analyses we analysed mothers and non-mothers together.

### Rater agreement on specific videos

Next, we tested whether a higher proportion of participants agreed on the type of look the infant gave in each video clip than expected by chance (0.33). A binomial test (0.33) was conducted for the type of look chosen by the highest number of raters for each video clip, to see if a higher proportion of raters than expected by chance chose that type of look. As we conducted 30 binomial tests on Set 1 and Set 2 of the videos we bonferroni-adjusted the significance threshold to 0.0016 (.05 / 30 binomial tests) meaning that 20 or more raters had to be in agreement on a specific video in Set 1 or Set 2 to be significantly above the level expected by chance.

For Set 1, out of 30 videos, there were 9 videos where the raters agreed on one type of look significantly more than chance (high agreement, see Table 3). Of these videos, five of the looks were rated as "sharing looks", two were rated as a "checking look", and two were rated as an "orienting look" by most raters. When we compared this to how the mothers rated their own videos, the mothers agreed with only 2/5 of the sharing looks (the other three were "unknown", "checking", and "orienting" according to the mothers); 1/2 of the checking looks (the other was rated "sharing" by the mother); and 0/2 of the two orienting looks (rated "unknown" and "checking" by the mother; S1 Table). This means that of the 26 looks mothers had classified as sharing, checking or orienting (excluding the 4 looks mothers rated as 'unknown'), in only 3 of these cases did a significant majority of raters agree with the categorisation of the mother.

For Set 2, out of 30 videos, there were 14 videos where the raters agreed on one look significantly more than chance (high agreement, see Table 3). Of these videos, six of the looks were rated as "sharing looks", seven were rated as "checking looks", and one "orienting look" by most raters.

We also tested whether raters were more or less confident for high agreement looks (as determined by the binomial tests; n = 23) or low agreement looks (where the two highest look types had a difference of ≤3 responses; n = 16). On a scale of 1 to 10, the mean confidence for high agreement looks was 6.19 (SE = 0.21) and for low agreement looks was 5.64 (SE = 0.20), and overall participants use a limited range of the scale (min 4.8, max 7.8). A paired t-test revealed that raters were significantly more confident about high agreement looks than low agreement looks ($t(31) = 6.72$, $p<0.001$).

## How are raters identifying high-agreement Sharing and Checking looks?

To understand the cues that raters may be relying on to make their judgements, we compared behaviour of the infant and mother coded from 11 videos that most raters agreed were Sharing Looks to the 9 videos that most raters agreed were Checking Looks (from the binomial analysis). We excluded the 3 videos where most raters agreed the infant look was an Orienting look, due to the low number. We examined whether high agreement Sharing and Checking looks differed in terms of (a) the duration of look from the infant to their mother's face; (b) the presence of mutual gaze between mother and infant (e.g. the mother reciprocating the infant's look to her face); (c) the presence/absence of (i) facial expressions, (ii) gestures, and (iii) vocal communication from infant to mother during the infant's look to their mother; (d) frequency of any communicative signals produced by the infant in whole video; (e) the presence/absence of (i) facial expressions, (ii) gestures, and (iii) vocal communication from mother to the infant during the infant's look to their mother; (f) frequency of any communicative signals produced by the mother in whole video.

**(a) the duration of look from the infant to their mother's face.** A GLMM, with "Duration of infant look to mother" as the response variable (gamma distribution) and "Type of agreed look" (Sharing / Checking) as the independent variable and "Video Dyad ID" as a random factor, confirmed that looks judged as 'sharing' were significantly longer than looks judged as 'checking' looks ($b = 0.84$, SE = 0.31, $p = 0.007$; Fig 1).

**(b) the duration of mutual gaze between mother and infant.** There were only 9 videos with mutual gaze (7 judged to be sharing looks; 2 judged to be checking looks), so the sample of mutual gaze occurrences was too small for inferential statistics to be appropriate. Descriptively, when mutual gaze was present it seemed to be for a longer duration in looks judged as 'sharing' (mean = 2.31s; SE = 0.52) than looks judged as 'checking' (mean = 0.53s; SE = 0.05).

**Table 3. Distribution of participant responses over the three categories of looks for the 23 looks identified as 'high agreement' by binomial tests.**

| Set | Mother's Answer | Number of Participants who selected 'Sharing' | Number of Participants who selected 'Checking' | Number of Participants who selected 'Orienting' |
|---|---|---|---|---|
| Set 1 | Sharing (+) | 24, p<0.001 | 2 | 6 |
| Set 1 | Checking (-) | 21, p<0.001 | 5 | 5 |
| Set 1 | Orienting (-) | 21, p<0.001 | 11 | 0 |
| Set 1 | Sharing (+) | 29, p<0.001 | 0 | 2 |
| Set 1 | Unknown | 26, p<0.001 | 4 | 2 |
| Set 1 | Checking (+) | 1 | 23, p<0.001 | 6 |
| Set 1 | Unknown | 0 | 4 | 27, p<0.001 |
| Set 1 | Checking (-) | 2 | 7 | 23, p<0.001 |
| Set 1 | Sharing (-) | 3 | 20, p<0.001 | 9 |
| Set 2 | – | 21, p<0.001 | 3 | 8 |
| Set 2 | – | 20, p<0.001 | 4 | 8 |
| Set 2 | – | 30, p<0.001 | 0 | 1 |
| Set 2 | – | 3 | 28, p<0.001 | 0 |
| Set 2 | – | 5 | 23, p<0.001 | 2 |
| Set 2 | – | 1 | 26, p<0.001 | 3 |
| Set 2 | – | 3 | 29, p<0.001 | 0 |
| Set 2 | – | 21, p<0.001 | 7 | 4 |
| Set 2 | – | 21, p<0.001 | 2 | 8 |
| Set 2 | – | 2 | 27, p<0.001 | 3 |
| Set 2 | – | 0 | 10 | 22, p<0.001 |
| Set 2 | – | 22, p<0.001 | 7 | 3 |
| Set 2 | – | 1 | 20, p<0.001 | 10 |
| Set 2 | – | 2 | 29, p<0.001 | 1 |

For new video trials, the mothers' answers are recorded as well as whether they agreed (+), or disagreed (-) with the raters, or whether their ratings were unknown. The type of look selected by the highest number of participants has been underlined.

(c) the presence/absence of (i) facial expressions, (ii) gestures, and (iii) vocal communication produced by the infant during their look to their mother. We then assessed whether

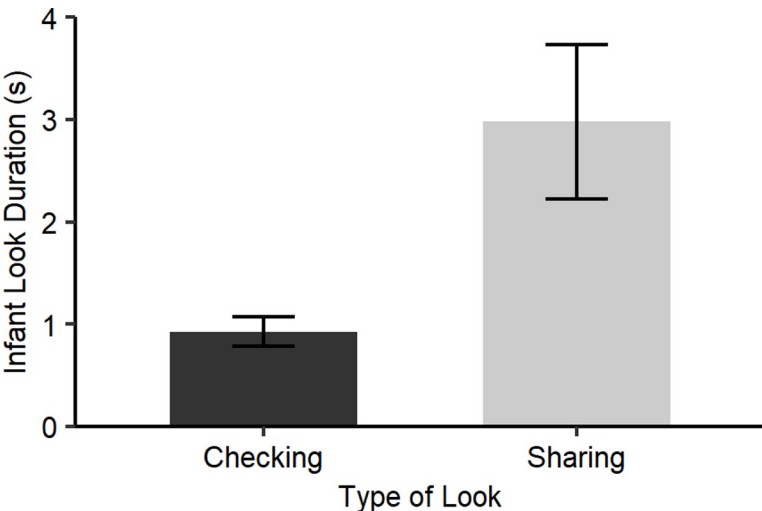

**Fig 1. Mean duration of look from infant to mother for 'high agreement' looks that were rated as "Checking looks" and "Sharing looks" by the majority of participants, with SEM error bars.**

the presence or absence of infant facial expressions, gestures, or vocal communication during an infant's look to their mother differed for high agreement Sharing and Checking looks. We ran three GLMMs with "Presence/Absence of [communication type]" as the response variable (binomial distribution), and the independent variable was "Type of agreed look" (Sharing / Checking) with "Video Dyad ID" as a random factor (Table 4). Although descriptively more communicative signals were produced by the infant during Sharing looks than Checking looks, the models indicated that there was no significant difference in the likelihood of infant communicative signals occurring in looks judged to be 'Sharing' and 'Checking'.

**(d) the presence/absence of (i) facial expressions, (ii) gestures, and (iii) vocal communication produced by the mother during the infant's look to their mother.** We then assessed whether the presence or absence of mother facial expressions, gestures, or vocal communication during an infant's look to their mother differed for high agreement Sharing and Checking looks. There was zero or highly limited variation in at least one of the categories of looks for each of the communication behaviours (Table 5), rendering inferential statistics inappropriate. Descriptively, for all three types of communication, there is almost no mother communication present for high agreement Checking looks, while there is more communication present in high agreement Sharing looks, with 100% of these having vocal communication present.

## Are raters responding to "charismatic" (i.e. more communicative) mothers or infants?

We also wanted to test whether raters attended to the overall rates of communication from the mothers and infants in the whole video clip, and perhaps used the heuristic that more

**Table 4. The proportion of high agreement Sharing and Checking looks (as judged by the raters) in which the infant produced at least one facial expression, gesture, and vocalisation during the look to their mother, and results from the GLMM analyses.**

|  | Sharing | Checking | GLMM for Presence/Absence of communication type |
| --- | --- | --- | --- |
| **Facial Expression Present** | 8/11 (0.73) | 3/9 (0.33) | $b = 2.17$, SE = 1.80, p = 0.229 |
| **Gesture Present** | 4/11 (0.36) | 1/9 (0.11) | $b = 8.83$, SE = 6.34, p = 0.164 |
| **Vocal Communication Present** | 7/11 (0.64) | 5/9 (0.56) | $b = 0.24$, SE = 0.92, p = 0.714 |

**Table 5. The proportion of high agreement Sharing and Checking looks (as judged by the raters) in which the mother produced at least one facial expression, gesture, or vocal communication during their infant's look to them.**

|  | Sharing | Checking |
|---|---|---|
| **Facial Expression Present** | 4/11 (0.36) | 0/9 (0.00) |
| **Gesture Present** | 6/11 (0.55) | 1/9 (0.11) |
| **Vocal Communication Present** | 11/11 (1.00) | 0/9 (0.00) |

communicative / charismatic individuals were more likely to share looks. To test this idea, we first ran a GLMM with "Frequency of mother communication" as the response variable (Gaussian distribution), "Type of agreed look" (Sharing / Checking) as the independent variable and "Video Dyad ID" as a random factor. Mother communication, in terms of the total number of facial expressions, gestures, and vocalisations combined, was significantly more frequent in high agreement videos rated as Sharing looks than those rated as Checking looks ($b$ = 2.06, SE = 0.43, p<0.001; Fig 2). When we ran the same GLMM but with "Frequency of infant communication" as the response variable, we did not see the same pattern, rather infant communication occurs at similar frequencies throughout the high agreement videos rated as Checking (mean = 1.667, SE = 0.333) and Sharing looks (mean = 2.545, SE = 0.312; $b$ = 0.809, SE = 0.47, p = 0.100).

## Discussion

As joint attention is so pivotal in human infant development and potentially also in human evolution, our main objective in this study was to determine whether the quality of an infant's look to their mother can be reliably identified and therefore is sufficient for distinguishing whether joint attention has occurred. We found overall low agreement among naïve raters in assigning looks from infants to their mothers as Sharing, Checking, or Orienting looks, suggesting that the definitions given by Hobson and Hobson were not adequate for our raters to be able to reliably assign these types of looks in infants [47]. Whilst both our study and Hobson and Hobson's study used naïve raters to identify looks from videos of dyadic interactions, there are several key differences between our studies which may explain our failure to replicate

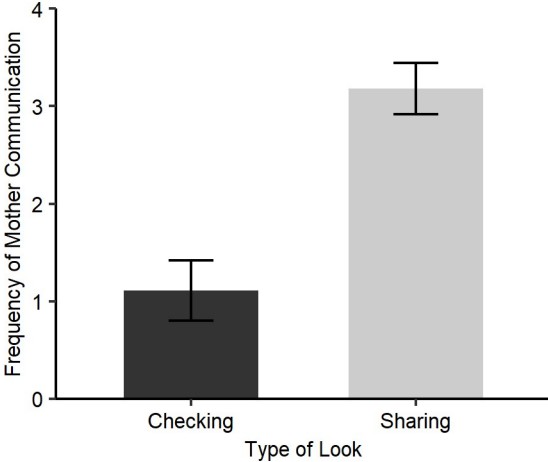

**Fig 2. Mean frequency of mother communication in high agreement looks rated by the majority of participants as "Checking looks" and "Sharing looks", with SEM error bars.**

their findings. Firstly, Hobson and Hobson used footage of 11-year-olds with and without autism interacting with an experimenter. Whilst they report that "mostly it was straightforward to 'feel' (and judge) whether the looks seen on videotape were sharing, expressive of interpersonal engagement; checking, indicative of glancing up to check the tester's face for a reaction or instruction; or orientating" (p. 419), looks from preverbal infants proved much more difficult to judge for our raters. Cues as to the intention underlying looks in preverbal infants may be subtler and more difficult to distinguish than in the older children in Hobson and Hobson's study.

There are also other important methodological differences between the studies: our raters were constrained to viewing a short video clip once at normal speed, whereas Hobson and Hobson don't specify any constraints on how much video surrounding the looks was reviewed, how many times raters viewed the videos, or at what speed. Second, whereas Hobson and Hobson had 2 raters assess just 27 looks from 6 children, we had 30 raters assess 60 looks from 15 infants, providing a more representative sample of both raters and looks. Whilst these differences mean that our study does not challenge Hobson and Hobson's original findings, our results question whether they can be extended to preverbal infants. In our controlled experiment, Sharing looks could not be reliably identified by naïve third-party observers in preverbal infants, suggesting that types of look should not be considered as a valid way of identifying joint attention events in preverbal infants.

We found not only that naïve observers had low levels of agreement with each other, but critically they rarely agreed with the judgement of the mother who received the looks. When considering the 26 looks that mothers categorised as sharing, checking or orienting, in only 3/26 cases did a significant majority of raters agree with the mothers' judgement of the type of look they received from their infants. It is possible that when engaged in an interaction and receiving a look directly, an individual can accurately infer the intentions of their interaction partner, and we hoped that the mothers' judgements made immediately after the play interaction would be an approximation of this experience. Our data shows clearly, however, that third party observers seem unable to access the same cues that the mothers received directly from their infants. This is perhaps not surprising as joint attention arises when partners dynamically perceive, interpret, and respond to behaviours in each other. As researchers, we cannot access a mother's experience of being involved in an interaction with her infant. Perhaps future research could assess if types of look can be reliably identified by multiple observers from footage obtained from head mounted cameras on the interaction partners. Whilst this doesn't replicate the experience of being the interaction partner, it may give a better approximation of the mothers' perspective of their interaction with their infant whilst still allowing the reliability of judgements to be assessed, which is fundamental to replicable, reliable scientific investigation.

We considered that people with more experience with infants may be more reliable at assigning the types of look (i.e. have higher levels of agreement), so we examined the responses of mothers and non-mothers separately. However, we found that mothers and non-mothers showed similar low levels of agreement on the type of look. Experience with young infants does not appear to make raters more reliable in coding Sharing, Checking, and Orienting looks. Another interesting finding was that raters were significantly more confident for high agreement looks compared to low agreement looks. We don't know the extent to which participants explicitly or implicitly relied on the behavioural cues that we coded in the videos, but given the positive relationship between a rater's self-report of confidence in their judgement and agreement between raters, future research may be able to usefully ask raters to reflect on the cues they relied on.

Taken together, our results indicate that in our study naïve third party observers were unable to categorise most infant looks reliably. Future research could investigate if reliability

can be improved by changing parameters of the current study, such as raters receiving more training before completing the task or raters watching longer clips that contain more contextual information, more times or in slow motion. Until robust evidence of parameters that may support reliable identification of look type by 3ʳᵈ party observers is available, we suggest that type of look should not be used to assess whether joint attention events have occurred in infants. Hobson and Hobson was the only paper we found that made explicit claims about the quality of looks as valid markers of joint attention [47], but there are many papers that implicitly use quality of looks to determine whether children "acknowledge their partner's participation" (Table 1), and subsequently whether joint attention has occurred. While the frequency and duration of looks between partners can be readily and objectively extracted from an interaction through video coding, our low levels of agreement on the types of look suggest that rating the quality of looks could be an unreliable way of assessing whether partners understand that they are both attending to the same object/event together and therefore engaging in a joint attention event.

We identified a subset of looks where significantly more raters than expected by chance agreed on the type of look as 'Checking' or 'Sharing' (only 3 looks were agreed to be Orienting looks, which was too few to analyse). Video coding of infant and mother behaviour during these high agreement videos revealed several potential behaviour cues that raters may have been relying on to distinguish these types of look. 'Sharing' looks were significantly longer than 'Checking' looks and, descriptively, were more likely to contain mutual gaze, and when it did occur, mutual gaze periods seemed to be longer. Gaze, attention shifts, gaze alternation, and mutual gaze are already used across many definitions of joint attention events (Table 1), and so including a measure of duration for looks and mutual gaze would be an objective and readily code-able variable to examine in future studies on joint attention.

When we looked at communication, there were no significant effects of infant communication on how the raters assigned looks. Because we had instructed the raters to attend to the infant's look to the mother, we expected that they would rely more on the infant's behaviour than the mother's. Surprisingly, we found the mother's but not the infant's overall rates of communication varied significantly with the classification of high agreement 'Sharing' and 'Checking' looks, indicating this may have been a cue the raters relied on to make their judgements. Taken together, it is possible that in the minority of cases where raters agreed at above chance rates, they may have been operating with the heuristics that longer or mutual looks are indicative of a sharing intent and that more communicative mothers are more likely to receive sharing looks from their infant. Whilst duration of look is a relevant cue, using the mother's communicativeness to make judgements about the infant's intentions is unlikely to be a reliable cue. This highlights the limitations of asking raters to make holistic judgements of stimuli: it is impossible to know which behavioural cues raters relied on at either an explicit or implicit level, and if raters are relying upon irrelevant cues their judgements would be invalid. We therefore advocate focussing on objective behavioural criteria for assessing sharing intentions, rather than rating subjective characteristics of behaviour.

Our study was conducted in a single species and in a single cultural context, providing favourable conditions for reliable judgements to be made, but our raters still failed to agree on the majority of their judgements. If we are to fully understand the ontogeny of joint attention we need to address the persistent sampling bias in developmental psychology for W.E.I.R.D. (Western, Educated, Industrialized, Rich, and Democratic) samples [113], and study early joint attention in diverse cultural contexts. Equally, in order to understand the evolutionary origins of joint attention we need to examine interactions in other species. As our results cast doubt on whether types of look can be reliably determined within a single species and cultural context, it seems highly unlikely they could be reliably identified in cross-cultural or cross-

species comparisons. Whilst subjective holistic judgements of behavioural intentions can seem appealing in that they may have the potential to capture nuanced, subtle cues that quantitative measurement of behaviour via video coding may miss, our study indicates that we need to move away from subjective approaches that assess the quality of looks and engagement based on a holistic viewing of an interaction. Moving towards an approach where the 'jointness' of a joint attention event and awareness of attending to the object/event together is based on extraction of detailed behaviours from videos, such as look duration, mutual gaze duration, and communication by both interactants, will yield conservative, but importantly reliable identification of joint attention events that can be applied across cultures and species. Looking ahead, the implementation of rigorous, reliable measures of behaviour will be crucial for understanding the ontogenetic and evolutionary origins of joint attention.

## Supporting information

**S1 Table. Confusion matrix between how mothers assigned their infants' looks and how raters assigned the looks for which they had high agreement.**
(DOCX)

**S1 Data.**
(XLSX)

**S1 File.**
(TXT)

## Acknowledgments

Many thanks to the mothers and infants who participated in this study; to Junior Whiteley who assisted in collecting and cutting the videos used; and to Yujin Lee for help with early versions of the video coding scheme.

## Author Contributions

**Conceptualization:** Kirsty E. Graham, Joanna C. Buryn-Weitzel, Nicole J. Lahiff, Claudia Wilke, Katie E. Slocombe.

**Data curation:** Kirsty E. Graham, Claudia Wilke.

**Formal analysis:** Kirsty E. Graham, Claudia Wilke.

**Funding acquisition:** Katie E. Slocombe.

**Investigation:** Kirsty E. Graham, Joanna C. Buryn-Weitzel, Nicole J. Lahiff, Katie E. Slocombe.

**Methodology:** Kirsty E. Graham, Claudia Wilke, Katie E. Slocombe.

**Project administration:** Kirsty E. Graham, Katie E. Slocombe.

**Resources:** Katie E. Slocombe.

**Software:** Katie E. Slocombe.

**Supervision:** Katie E. Slocombe.

**Validation:** Claudia Wilke.

**Visualization:** Kirsty E. Graham.

**Writing – original draft:** Kirsty E. Graham, Joanna C. Buryn-Weitzel, Nicole J. Lahiff, Claudia Wilke, Katie E. Slocombe.

**Writing – review & editing:** Kirsty E. Graham, Joanna C. Buryn-Weitzel, Nicole J. Lahiff, Claudia Wilke, Katie E. Slocombe.

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
