## [Decision Letter · Decision Letter 0]

13 Apr 2021

PONE-D-21-06287

Detecting joint attention events in mother-infant dyads: sharing looks cannot be reliably identified

PLOS ONE

Dear Dr. Graham,

Thank you for submitting your manuscript to PLOS ONE. After careful consideration, we feel that it has merit but does not fully meet PLOS ONE’s publication criteria as it currently stands. Therefore, we invite you to submit a revised version of the manuscript that addresses the points raised during the review process.

Both reviewers and I found that your paper has merit. However, they highlighted a couple of points that I would like you to take into account. Particularly, Reviewer #2 raised some significant theoretical problems. Please, consider all the Reviewers' comments and suggestions.

We look forward to receiving your revised manuscript.

Kind regards,

Ewa Pisula

Academic Editor

PLOS ONE

Journal Requirements:

In line with PLOS' guidelines on detailed reporting (https://journals.plos.org/plosone/s/criteria-for-publication#loc-3), please ensure that you have provided sufficient detail on the children/infants in the Methods section, including their ages.

Please include captions for your Supporting Information files at the end of your manuscript, and update any in-text citations to match accordingly. Please see our Supporting Information guidelines for more information: http://journals.plos.org/plosone/s/supporting-information.

Reviewers' comments:

Reviewer's Responses to Questions

**Comments to the Author**

1. Is the manuscript technically sound, and do the data support the conclusions?

Reviewer #1: Partly

Reviewer #2: Partly

2. Has the statistical analysis been performed appropriately and rigorously? 

Reviewer #1: Yes

Reviewer #2: I Don't Know

3. Have the authors made all data underlying the findings in their manuscript fully available?

Reviewer #1: Yes

Reviewer #2: No

4. Is the manuscript presented in an intelligible fashion and written in standard English?

Reviewer #1: Yes

Reviewer #2: Yes

5. Review Comments to the Author

Reviewer #1: Thank you for the opportunity to review this manuscript. The research concerns an important problem of the reliability of the assessment of joint attention (sharing looks) by observers. The results indicated that sharing looks cannot be reliably assessed if the assessment is based on the operationalization proposed by Hobson and Hobson (2007).

Title

I feel that the title is too strong because it implies that sharing looks cannot be reliably assessed at all, whereas the results indicate that sharing looks cannot be reliably identified while following Hobson and Hobson’s (2007) operationalization and one specific coding procedure (the one employed by the Authors), but not all possible operationalizations and procedures. I suggest attenuating the claim in the title.

Abstract

The abstract accurately summarizes the study.

Introduction

The introduction is thorough and well-written. Table 1 is very helpful.

Page 5, line 82: Simply looking at the same object at the same time can also be called parallel attention (Gaffan, Martins, Healy, & Murray, 2010), which is an earlier developmental milestone than joint attention.

Page 6, lines 100-103. Please, provide a reference (suggested by who?);

line 104 “acknowledges partner’s participation” – as this is a quote, please provide a reference with page number.

Method

Page 9, line 141: I am wondering whether the judgements would be more accurate if the videos had been longer (if participants had been able to see them in a broader context). Perhaps this is something that could be discussed later on. Similarly, perhaps the reliability could be higher if the judges had been able to view the videos more than once or in slow motion.

Results

Page 13, line 247: There is a comma where a full stop should be.

Overall: Did the Authors compute an agreement between themselves in identifying sharing looks? Were there any analyses conducted that involved an assessment of the agreement between skilled coders (e.g., trained undergrads/research assistants)?

Discussion

Page 18, line 359: Maybe specify that this concerns infants.

Overall: I feel that the conditions of the assessment, that is only being able to view very short vignettes once, made the task very hard for participants. In the context of scientific research, coders are usually able to replay videos, also in slow motion. Furthermore, training sessions are provided, and not just definitions. Therefore, the conclusion that sharing looks cannot be reliably identified based on holistic judgements is too strong.

Reviewer #2: Thankyou for the opportunity to review this paper, which represents an interesting analysis of the use of gaze as an indicator of joint attention. The authors rightly identify joint attention as a significant learning context for young children, and focus on the operationalization of this context. In particular, they focus on whether the assessment of different gaze functions, as specified by Hobson and Hobson (2007) can be reliably used to indicate mother-infant joint attention to a third party.

While the authors have developed and reported on an innovate study to attempt to address this question, my feedback on the manuscript is more conceptual than methodological. I will summarise below:

1. Firstly, the context of comparison between this study and the one by Hobson and Hobson is vast, with the H&H paper involving looks between autistic or developmentally delayed children and an experimenter, while this study involves looks between mothers and infants. I would expect that the behavioural indicators of the latter would be much more subtle than the former, so do not feel that a firm conclusion can be made about whether or not gaze alone can be reliably assessed.

2. Joint attention with young infants is an inherently relationship-based context, where each participant perceives, interprets and responds to the cues of the other. For this reason, I am not surprised that observers who did not know the mother or infant could not reliably judge the type of the gaze. For example, it is difficult to know whether an infant is looking deep into the partner's eyes unless you are, in fact, that partner and are experiencing this.

3. Related to the above, I am also not surprised that the participants found it difficult to achieve any degree of reliability in their assessment. As noted in Table 1, the majority of research articles on joint attention use a range of cues to determine whether or not a state of joint attention is achieved. Most of these studies would include the training of observers and raters that goes beyond the provision of a definition provided in this paper. The finding that particular agreed looks were accompanied by other behavioural cues such as length of gaze and communicative overtures suggests that untrained observers intuitively focus on more than simple gaze.

I am afraid that these points above leave me to question the implications and overall significance of the findings for understanding the ontogenetic and evolutionary origins of joint attention. Instead, they highlight to me that joint attention is a mutually constituted interactional context which is a product of a dynamic and 'in the moment' combination of skills and behaviours from both interacting parties. This is touched on at the end, but if revised, I would expect that a stronger critique and interpretation is included in the discussion, which at present, largely summarises but does not explicitly interpret or explain the findings. Without this, the discussion of cross cultural and cross species comparisons seems quite out of place.

A final comment relates to the accessibility of the data, which does not seem to comply with requirements. The author has stated that all data is in the paper, but my reading of requirements is that the full data set should be accessible in a repository such as Open Science. I cannot see from the comments that this is the case.

6. PLOS authors have the option to publish the peer review history of their article (what does this mean?). If published, this will include your full peer review and any attached files.

Reviewer #1: **Yes: **Alicja Niedźwiecka

Reviewer #2: No

---

## [Author Response · Author response to Decision Letter 0]

31 May 2021

Response to Reviewers

We thank the reviewers for their helpful comments. We have addressed each point raised below and are confident that the manuscript is greatly improved as a result of these changes. All line numbers refer to the track change version of the revised manuscript. We have also updated the manuscript with the formatting requirements shared with us by the editor.

Reviewer #1: "Thank you for the opportunity to review this manuscript. The research concerns an important problem of the reliability of the assessment of joint attention (sharing looks) by observers. The results indicated that sharing looks cannot be reliably assessed if the assessment is based on the operationalization proposed by Hobson and Hobson (2007)."

Thank you! We have addressed each of your comments below.

"Title

I feel that the title is too strong because it implies that sharing looks cannot be reliably assessed at all, whereas the results indicate that sharing looks cannot be reliably identified while following Hobson and Hobson’s (2007) operationalization and one specific coding procedure (the one employed by the Authors), but not all possible operationalizations and procedures. I suggest attenuating the claim in the title."

We have updated the title to provide more clarification about the conditions: “Detecting joint attention events in mother-infant dyads: sharing looks cannot be reliably identified by naïve third-party observers”

"Abstract

The abstract accurately summarizes the study."

Thank you – in line with the modification of the title we have now clarified that the raters in both our study and the original Hobson and Hobson paper were naïve raters.

"Introduction

The introduction is thorough and well-written. Table 1 is very helpful."

Thank you!

"Page 5, line 82: Simply looking at the same object at the same time can also be called parallel attention (Gaffan, Martins, Healy, & Murray, 2010), which is an earlier developmental milestone than joint attention."

Thank you, this is very important to note. We have now added clarification of the other term used to describe this behaviour (line 84): 

“At the simplest end, it is sufficient for two individuals to both look at the same object (also called ‘parallel attention’ (39)), or for two individuals to look at one another; and at the more complex end, individuals must coordinate their attention between an object or event and one another and also communicate with one another about it (Table 1).”

And we have noted at the end of the paragraph that given the differences in developmental trajectories indicates different cognitive processes may underlie different behaviours currently labelled joint attention (line 96):

 “It is also important to note that simpler forms of ‘joint attention’ such as the simultaneous monitoring of an object or event by multiple observers emerge earlier in development (common in 6-month olds (39)) than more complex forms of joint attention, supporting the idea that different cognitive processes may underlie some behaviours currently labelled as ‘joint attention’ in the literature.”

"Page 6, lines 100-103. Please, provide a reference (suggested by who?);

line 104 “acknowledges partner’s participation” – as this is a quote, please provide a reference with page number."

We cite the table in the text at this point as many studies use this definition, but as suggested have also provided an example of a reference using this definition with the relevant page number “(Table 1; e.g. (13), p. 19)” (line 109)

"Method

Page 9, line 141: I am wondering whether the judgements would be more accurate if the videos had been longer (if participants had been able to see them in a broader context). Perhaps this is something that could be discussed later on. Similarly, perhaps the reliability could be higher if the judges had been able to view the videos more than once or in slow motion."

In terms of the length of videos, we wanted to test whether people could judge the look itself, as this is what the Hobson & Hobson paper claims to be assessing. We agree that it is likely that with longer videos with more context there may have been more agreement, but this would suggest that the raters were using context or behaviours alongside the looks to assess the type rather than the look itself. Indeed, even with clips that were this short, there was a very strong effect of the general communicativeness of the mother on how raters assigned looks (which is separate from the actual look itself). 

"On line 150, we have now added our rationale for this time-window:

 “This time window was chosen to give raters an opportunity to observe part of the interaction in which the look occurred, but without providing broader context and behaviours, as we wanted raters to judge the quality of the look, not the general intent of the infant or mother.”"

In terms of repeating the video or being able to view it in slow motion, these are interesting ideas that future research could test. The rationale for opting for a single viewing in our experiment was that we wanted to standardise the information the participants had to make their judgements on (we did not think we could fairly compare the ratings of a participant who watched all the clips once, with a participant who chose to re-watch some clips multiple times) and we thought fatigue might adversely affect performance if we made multiple viewings before judgement compulsory for each look. 

We now offer an explanation of this rationale (line 187):

“We wanted to both standardise the number of times participants could view a video and to reduce the effects of fatigue on performance. Therefore, the video played only once in each trial.”

We have also added a section to the discussion highlighting that future research may usefully search for parameters which might support the reliable identification of look type (line 419-425):

 “Taken together, our results indicate that in our study naïve third party observers were unable to categorise most infant looks reliably. Future research could investigate if reliability can be improved by changing parameters of the current study, such as raters receiving more training before completing the task or raters watching longer clips that contain more contextual information, more times or in slow motion. Until robust evidence of parameters that may support reliable identification of look type by 3rd party observers is available, we suggest that type of look should not be used to assess whether joint attention events have occurred in infants.”

"Results

Page 13, line 247: There is a comma where a full stop should be."

Thanks for catching that. We have fixed it (now line 260).

"Overall: Did the Authors compute an agreement between themselves in identifying sharing looks? Were there any analyses conducted that involved an assessment of the agreement between skilled coders (e.g., trained undergrads/research assistants)?"

No, we did not compute agreement between authors on these looks and all our raters were naïve participants. We now make it clear that our results are based on the judgements of naïve raters in the title, abstract and discussion. 

"Discussion

Page 18, line 359: Maybe specify that this concerns infants."

We have now specified ‘infant development’ (line 366)

"Overall: I feel that the conditions of the assessment, that is only being able to view very short vignettes once, made the task very hard for participants. In the context of scientific research, coders are usually able to replay videos, also in slow motion. Furthermore, training sessions are provided, and not just definitions. Therefore, the conclusion that sharing looks cannot be reliably identified based on holistic judgements is too strong."

Thank you for highlighting these important points. We are now more careful to constrain our discussion to the parameters of the study we conducted and be more tentative with our conclusions throughout. We have also introduced a new section in the discussion to explicitly acknowledge your points that training, length of video, multiple viewings and slow motion viewing may yield different results (line 420)

“Future research could investigate if reliability can be improved by changing parameters of the current study, such as raters receiving more training before completing the task or raters watching longer clips that contain more contextual information, more times or in slow motion. Until robust evidence of parameters that may support reliable identification of look type by 3rd party observers is available, we suggest that type of look should not be used to assess whether joint attention events have occurred in infants.”

"Reviewer #2: Thankyou for the opportunity to review this paper, which represents an interesting analysis of the use of gaze as an indicator of joint attention. The authors rightly identify joint attention as a significant learning context for young children, and focus on the operationalization of this context. In particular, they focus on whether the assessment of different gaze functions, as specified by Hobson and Hobson (2007) can be reliably used to indicate mother-infant joint attention to a third party.

While the authors have developed and reported on an innovate study to attempt to address this question, my feedback on the manuscript is more conceptual than methodological. I will summarise below:

1. Firstly, the context of comparison between this study and the one by Hobson and Hobson is vast, with the H&H paper involving looks between autistic or developmentally delayed children and an experimenter, while this study involves looks between mothers and infants. I would expect that the behavioural indicators of the latter would be much more subtle than the former, so do not feel that a firm conclusion can be made about whether or not gaze alone can be reliably assessed."

We agree that more discussion concerning the differences between our study and the H&H paper is needed and we have now inserted a section into the discussion to suggest some of the points you raise above and be clearer that we are questioning whether their findings can be extended to preverbal infants, rather than challenging their original findings (lines 372-391)

 “Whilst both our study and Hobson and Hobson’s study used naïve raters to identify looks from videos of dyadic interactions, there are several key differences between our studies which may explain our failure to replicate their findings. Firstly, Hobson and Hobson used footage of 11-year-olds with and without autism interacting with an experimenter. Whilst they report that “mostly it was straightforward to ’feel’ (and judge) whether the looks seen on videotape were sharing, expressive of interpersonal engagement; checking, indicative of glancing up to check the tester’s face for a reaction or instruction; or orientating” (p. 419), looks from preverbal infants proved much more difficult to judge for our raters. Cues as to the intention underlying looks in preverbal infants may be subtler and more difficult to distinguish than in the older children in Hobson and Hobson’s study.

There are also other important methodological differences between the studies: our raters were constrained to viewing a short video clip once at normal speed, whereas Hobson and Hobson don’t specify any constraints on how much video surrounding the looks was reviewed, how many times raters viewed the videos, or at what speed. Second, whereas Hobson and Hobson had 2 raters assess just 27 looks from 6 children, we had 30 raters assess 60 looks from 15 infants, providing a more representative sample of both raters and looks. Whilst these differences mean that our study does not challenge Hobson and Hobson’s original findings, our results question whether they can be extended to preverbal infants. In our controlled experiment, Sharing looks could not be reliably identified by naïve third-party observers in preverbal infants, suggesting that types of look should not be considered as a valid way of identifying joint attention events in preverbal infants.”

"2. Joint attention with young infants is an inherently relationship-based context, where each participant perceives, interprets and responds to the cues of the other. For this reason, I am not surprised that observers who did not know the mother or infant could not reliably judge the type of the gaze. For example, it is difficult to know whether an infant is looking deep into the partner's eyes unless you are, in fact, that partner and are experiencing this."

We also agree with this point, and have clarified in the title that our study refers to naïve third-party observers. As researchers, we cannot take the second-person perspective in these interactions and so it is crucial to develop a toolkit of clearly operationalised behaviours we can code reliably from observed interactions. We have added this section to the discussion to address this point (line 399):

“Our data shows clearly, however, that third party observers seem unable to access the same cues that the mothers received directly from their infants. This is perhaps not surprising as joint attention arises when partners dynamically perceive, interpret, and respond to behaviours in each other. As researchers, we cannot access a mother’s experience of being involved in an interaction with her infant. Perhaps future research could assess if types of look can be reliably identified by multiple observers from footage obtained from head mounted cameras on the interaction partners. Whilst this doesn’t replicate the experience of being the interaction partner, it may give a better approximation of the mothers’ perspective of their interaction with their infant whilst still allowing the reliability of judgements to be assessed, which is fundamental to replicable, reliable scientific investigation.”

"3. Related to the above, I am also not surprised that the participants found it difficult to achieve any degree of reliability in their assessment. As noted in Table 1, the majority of research articles on joint attention use a range of cues to determine whether or not a state of joint attention is achieved. Most of these studies would include the training of observers and raters that goes beyond the provision of a definition provided in this paper. The finding that particular agreed looks were accompanied by other behavioural cues such as length of gaze and communicative overtures suggests that untrained observers intuitively focus on more than simple gaze."

We have now acknowledged that training and different viewing conditions may affect the reliability with which raters can judge types of look and that future research could examine this possibility (lines 420-425). Interestingly, one of H&H’s two judges is reported to be ‘naïve’ and training of judges is not mentioned in their paper, indicating that our failure to find reliable judgements is unlikely to be a simple product of our raters not receiving training prior to completing the task. 

"I am afraid that these points above leave me to question the implications and overall significance of the findings for understanding the ontogenetic and evolutionary origins of joint attention. Instead, they highlight to me that joint attention is a mutually constituted interactional context which is a product of a dynamic and 'in the moment' combination of skills and behaviours from both interacting parties. This is touched on at the end, but if revised, I would expect that a stronger critique and interpretation is included in the discussion, which at present, largely summarises but does not explicitly interpret or explain the findings. Without this, the discussion of cross cultural and cross species comparisons seems quite out of place."

We hope we have now addressed your concerns above and in particular, as requested we now provide a fuller discussion of the difference between second person and third person perspectives on interactions. We hope that it is now clear that we think our data is relevant to the development of clear, reliable and valid measures of joint attention, that are absolutely essential if we are to understand how joint attention emerges in humans from different cultures and the extent to which it might be present in non-humans. We are not questioning that joint attention arises in complex dynamic dyadic interactions, but we have to be able to reliably identify when this occurs from a third person perspective using objective, valid criteria. Our data suggested that subjective, holistic judgements are likely not a reliable way for third-party observers to identify joint attention events in preverbal infants. We hope that calling for the field to move forward in a way that makes our research reliable and replicable will be advantageous for us all and particularly those taking cross-cultural and comparative approaches to this topic. 

"A final comment relates to the accessibility of the data, which does not seem to comply with requirements. The author has stated that all data is in the paper, but my reading of requirements is that the full data set should be accessible in a repository such as Open Science. I cannot see from the comments that this is the case."

We included the data in supplementary materials, as this allows readers to easily access the data alongside the paper – and this is one of the recommended routes suggested by PLOS. Please can the editor advise if we have misunderstood the journal’s preferences and would rather the data be hosted elsewhere?

---

## [Decision Letter · Decision Letter 1]

13 Jul 2021

Detecting joint attention events in mother-infant dyads: sharing looks cannot be reliably identified by naïve third-party observers

PONE-D-21-06287R1

Dear Dr. Graham,

We’re pleased to inform you that your manuscript has been judged scientifically suitable for publication and will be formally accepted for publication once it meets all outstanding technical requirements.

Kind regards,

Ewa Pisula

Academic Editor

PLOS ONE

Additional Editor Comments (optional):

Dear Authors,

Both reviewers considered that all of their comments had been addressed in the revision of your paper. I find this paper very interesting and bringing valuable information for further research on joint attention in mother-infant dyads. Congratulations on your job!

Reviewers' comments:

Reviewer's Responses to Questions

**Comments to the Author**

1. If the authors have adequately addressed your comments raised in a previous round of review and you feel that this manuscript is now acceptable for publication, you may indicate that here to bypass the “Comments to the Author” section, enter your conflict of interest statement in the “Confidential to Editor” section, and submit your "Accept" recommendation.

Reviewer #1: All comments have been addressed

Reviewer #2: All comments have been addressed

2. Is the manuscript technically sound, and do the data support the conclusions?

Reviewer #1: (No Response)

Reviewer #2: (No Response)

3. Has the statistical analysis been performed appropriately and rigorously? 

Reviewer #1: (No Response)

Reviewer #2: (No Response)

4. Have the authors made all data underlying the findings in their manuscript fully available?

Reviewer #1: (No Response)

Reviewer #2: (No Response)

5. Is the manuscript presented in an intelligible fashion and written in standard English?

Reviewer #1: (No Response)

Reviewer #2: (No Response)

6. Review Comments to the Author

Reviewer #1: Thank you for addressing my comments. I appreciate your work. I have no further questions or concerns.

Reviewer #2: Thankyou for the opportunity to review the revision of this manuscript. I thank the authors for their close attention to my original review comments, and the revisions that have been made in response. I believe now that the interpretation of the data is much stronger and the focus is more clear than the original submission. The responses have addressed my feedback effectively.

7. PLOS authors have the option to publish the peer review history of their article (what does this mean?). If published, this will include your full peer review and any attached files.

Reviewer #1: **Yes: **Alicja Niedźwiecka

Reviewer #2: No

---

## [Editor Report · Acceptance letter]

15 Jul 2021

PONE-D-21-06287R1 

Detecting joint attention events in mother-infant dyads: sharing looks cannot be reliably identified by naïve third-party observers 

Dear Dr. Graham:

I'm pleased to inform you that your manuscript has been deemed suitable for publication in PLOS ONE. Congratulations! Your manuscript is now with our production department. 

Kind regards, 

on behalf of

Dr. Ewa Pisula 

Academic Editor

PLOS ONE